# SPEADI: Accelerated Analysis of IDP-Ion Interactions from MD-Trajectories

**DOI:** 10.3390/biology12040581

**Published:** 2023-04-10

**Authors:** Emile de Bruyn, Anton Emil Dorn, Olav Zimmermann, Giulia Rossetti

**Affiliations:** 1Jülich Supercomputing Centre, Forschungszentrum Jülich, 52425 Jülich, Germany; 2Faculty of Mathematics, Computer Science and Natural Sciences, RWTH Aachen University, 52062 Aachen, Germany; 3Computational Biomedicine, Institute for Advanced Simulation IAS-5 and Institute of Neuroscience and Medicine INM-9, Forschungszentrum Jülich, 52425 Jülich, Germany; 4Department of Neurology, RWTH Aachen University, 52062 Aachen, Germany

**Keywords:** intrinsically disordered proteins, MD simulation, ion distribution, ion dynamics, Alpha-Synuclein, humanin

## Abstract

**Simple Summary:**

Intrinsically Disordered Proteins (IDPs) are particularly sensitive to changes in chemical environmental conditions. Changes in this environment lead to alterations of their normal functions. We introduce the concept of a Time-Resolved Radial Distribution Function (TRRDF). TRRDFs are able to characterize the local environment dynamics in simulations around dynamically changing IDPs. TRRDFs are implemented and available in our open-source Python package SPEADI. We use SPEADI to characterize the dynamic distribution of ions around two IDPs Alpha-Synuclein (AS) and Humanin (HN) from Molecular Dynamics (MD) simulations. We analyze and explore the local ion–residue interactions that play an important role in the structures and behaviors of IDPs.

**Abstract:**

The disordered nature of Intrinsically Disordered Proteins (IDPs) makes their structural ensembles particularly susceptible to changes in chemical environmental conditions, often leading to an alteration of their normal functions. A Radial Distribution Function (RDF) is considered a standard method for characterizing the chemical environment surrounding particles during atomistic simulations, commonly averaged over an entire or part of a trajectory. Given their high structural variability, such averaged information might not be reliable for IDPs. We introduce the Time-Resolved Radial Distribution Function (TRRDF), implemented in our open-source Python package SPEADI, which is able to characterize dynamic environments around IDPs. We use SPEADI to characterize the dynamic distribution of ions around the IDPs Alpha-Synuclein (AS) and Humanin (HN) from Molecular Dynamics (MD) simulations, and some of their selected mutants, showing that local ion–residue interactions play an important role in the structures and behaviors of IDPs.

## 1. Introduction

The chemistry of biological molecules in aqueous solutions is influenced and modulated to a large degree by the chemical environment surrounding them. The presence and concentration of various ionic species in a solution have the effect of either adding or decreasing electronic repulsion as well as either increasing or decreasing the hydrophobic effect [1,2,3], which is among the main driving forces behind the folding of proteins [4,5]. The specific interaction between the surface residues of proteins and ions depends not only on the type of ion and amount of electrostatic screening in the solution, but also on the protein environment surrounding the residue [6,7].

The effect of ions is of significant importance to IDPs, a class of proteins that lack a stable tertiary structure under physiological conditions [8,9]. These proteins indeed possess a number of transient meta-stable states, and as such, IDPs cannot be described by a single or even two or three tertiary protein structures. Instead, they have to be characterized in terms of structural ensembles [10,11]. IDPs are sensitive to changes in solution conditions owing to their structural variability and primary protein structures containing higher fractions of charged and polar amino acid residues compared to globular proteins [12,13]. It has been shown that ion neutralization of local charge densities on IDPs’ surfaces can result in the preference of certain conformational states [14,15,16]. This is relevant, as IDPs are often folding to specific conformations when interacting with cellular partners [17]. They are also linked to pathological misfolded states in those suffering from neurological disorders [18,19,20], cancer [18,21,22], cardiovascular diseases [18,23], or diabetes [24,25]. As such, the study of their interaction with solvents and ions is of crucial importance to understanding their dynamics and characterization for possible pharmacological intervention [26,27].

Characterizing the local environment and interactions of atoms in biomolecular simulations with both solvents and other solutes is a challenging problem. Due to the large number of solvent molecules present in explicit solvent simulations, the equilibration of solvent positions and response to conformational changes in a biomolecule are fast compared to the time scale involved in conformational changes. In contrast, the equilibration of dilute solutes such as ions present in the solution is slow enough as to represent a rate-limiting step on the simulation itself [28].

To date, RDFs are commonly used to characterize the hydration shells of simulated biomolecules [29,30] and assess the structure and order of phospholipid bilayers [24,31]. The software currently available [32,33,34,35] to calculate RDFs of biomolecules provides an average over the whole or part of a simulation that has reached equilibrium. When the conformation of a protein is stable over the simulation time, this approach can provide a fairly accurate description of key solvent properties such as coordination numbers, mean distances, potentials of mean force and hydration [36,37,38,39]. The shape of a solvent’s RDF peaks even correlate with hydration dynamics [40].

For IDPs, which are characterized by several transient meta-stable states and many significantly different conformations at the equilibrium, average behaviors do not accurately reflect any of these conformational states. RDFs averaged over a simulation cannot identify key changes in a solution environment surrounding individual atoms that are relevant for an IDP’s function. For instance, coupled folding and binding reactions, where an IDP folds upon binding to its target protein, have been shown experimentally to be affected by the presence of different common salts beyond simple electrostatic effects, such that their affinities are ion-specific and occur at physiological concentrations [41].

Understanding and investigating such phenomena therefore requires a tool that provides time-resolved distributions sampled at different points on the protein surface. Unfortunately, computing RDFs is a computationally expensive task [42]. Additionally, storing separate RDFs for hundreds or thousands of time windows can also be memory-intensive.

We developed the tool SPEADI to meet this need, which provides an easy-to-use interface to calculate TRRDFs for any number of atoms in parallel while keeping a small memory footprint. Our tool exploits and scales with available computer hardware, from multiple CPUs to GPUs or even Google TPUs (Tensor Processing Units, custom hardware designed specifically for machine learning applications). Although SPEADI has been designed specifically with the characterization of ions interacting with IDPs in mind, it is agnostic towards atom type and simulation parameters; therefore, it can be used also for membrane components, as well as other biological systems. It can calculate RDFs, TRRDFs and van Hove Functions (VHFs) [43] for any atomistic simulation trajectory that the MDTraj [34] Python package can read.

Here, we apply our tool to the IDPs AS and HN, revealing that site-specific ion–residue interactions correlate with and inform on conformational and transient ion-stabilized states linked to their function.

## 2. Materials and Methods

### 2.1. Radial Distribution Function

The RDF describes the density of points surrounding a reference point. Applied to a system of particles, it describes the particle density surrounding a point as a function of distance. Fluctuations in the density are caused by microscopic inter-particle interactions and the RDF allows the extrapolation of macroscopic thermodynamic properties by approximating the equation of state [44,45]. The RDF is commonly normalized using bulk particle density:(1)$$\begin{array}{c} g\left(r\right)=\frac{1}{\rho_{\mathrm{bulk}}}\rho \left(r\right), \end{array}$$
so that values above 1 express an increased density or likelihood of finding particles compared to the bulk solution. RDFs find usage in simulations of biomolecules in a solution by quantifying the local structure surrounding atoms or sites of interest. For the derivation of the RDF for identical particles and between pairs of different types of particles, please refer to standard formulations of statistical thermodynamics [36].

The RDF showing how particles of type *b* are distributed around particles of type *a* during a simulation can be expressed as the mean of a binned counting function $dn_{ij}\left(r,t\right)$ over the trajectory length *T* and all pairs ij,
(2)$$\begin{array}{c} g_{a,b}\left(r\right)=\frac{1}{V\left(r\right)\rho_{b}^{\mathrm{bulk}}N_{a}N_{\mathrm{frames}}}\sum_{t=0}^{T}\sum_{i\in a}^{N_{a}}\sum_{j\in b}^{N_{b}}dn_{ij}\left(r,t\right) \end{array}$$
where $g_{a,b}\left(r\right)$ is normalized by $N_{a}$ and $N_{b}$ to account for the number of both types of particle in the simulation cell, as well as V(r), the volume of the radial shell, and $\rho_{b}^{\mathrm{bulk}}$, the bulk density of *b* particles. The counting function $n_{ij}$ equals either zero or one, depending on whether the distance between particles *i* and *j* is within the radial distance bin *r* with a bin size of d*r*,
(3)$$\begin{array}{c} dn_{ij}\left(r,t\right)=\delta \left(r-r_{i,j}\left(t\right)\right)=\left\{\begin{array}{cc} 0, & \mathrm{if}|r-r_{i,j}\left(t\right)|>dr\\ 1, & \mathrm{if}|r-r_{i,j}\left(t\right)|\leq dr \end{array}\right.\;, \end{array}$$
where $r_{i,j}\left(t\right)$ is the distance between particles *i* and *j* at time *t* during the simulation. The radial shell volume itself also depends on the radius: (4)$$\begin{array}{c} V\left(r\right)=\int_{r_{\mathrm{inner}}}^{r_{\mathrm{outer}}}4\pi r^{2}dr=\frac{4}{3}\pi \left(r_{\mathrm{outer}}^{3}-r_{\mathrm{inner}}^{3}\right). \end{array}$$

### 2.2. Time-Resolved Radial Distribution Function

For simulations of IDPs transitioning between many meta-stable states, expressing $g\left(r\right)$ as a mean over the whole simulation trajectory may obscure information regarding those states. To alleviate this problem, we introduce the concept of a Time-Resolved Radial Distribution Function (TRRDF) in SPEADI, where $g\left(r\right)$ is calculated as a mean over a discrete time window *W* with $N_{W}$ frames: (5)$$\begin{array}{c} g_{a,b}\left(r,W\right)=\frac{1}{V\left(r\right)\rho_{b}^{\mathrm{bulk}}N_{a}N_{W}}\sum_{t\in W}^{N_{W}}\sum_{i\in a}^{N_{a}}\sum_{j\in b}^{N_{b}}dn_{ij}\left(r,t\right). \end{array}$$

In this paper, time window will refer to the part of the overall trajectory that is selected to calculate one separate RDF. SPEADI allows the number of frames used to define a time window to be arbitrarily defined by the user, depending on the dimension of the system and on how frequently the trajectory was saved.

To calculate the TRRDF, a distance matrix $r_{ij}$ between each target group of particles and the reference group of particles is constructed for each frame in the window. A histogram of the distance matrix is then constructed and normalized according to the volume of each radial shell.

Multiplying $g\left(r\right)$ with the bulk density and radial shell volume then integrating over the distance gives the running coordination number $n\left(r\right)$ [36]: (6)$$\begin{array}{cc} n\left(r\right) & =V\left(r\right)\rho^{\mathrm{bulk}}\int_{0}^{r}g\left(r\right)dr \end{array}$$

This is simply the number of particles surrounding the reference point up to the distance *r*. $n\left(r\right)$ can be used to count the number of particles in the coordination sphere or hydration sphere of the particles of interest [36]. The coordination number between particle types *a* and *b* in time window *W*, $n_{a,b}\left(r,W\right)$ can be expressed as: (7)$$\begin{array}{cc} n_{a,b}\left(r,W\right) & =V\left(r\right)\rho_{b}^{\mathrm{bulk}}\int_{0}^{r}g_{a,b}\left(r,W\right)dr\\ \; & =\int_{0}^{r}\frac{1}{N_{a}N_{W}}\sum_{t\in W}^{N_{W}}\sum_{i\in a}^{N_{a}}\sum_{j\in b}^{N_{b}}dn_{ij}\left(r,t\right)dr. \end{array}$$

This enables calculating the running coordination number without defining a bulk density, which is useful if the target particles are part of a larger biomolecule or in simulations with changing volume.

In practice, values for $n_{a,b}\left(r,W\right)$, when looking at dilute solutes such as ions, are often fractional. This is because they move in and out of hydration shells during a time window. We therefore interpret these fractional values as the likelihood of finding a particle of type *b* within the radius *r* of particle type *a* within the time window *W*.

As SPEADI is able to efficiently compute TRRDFs in parallel for large numbers of sites, finding the time-resolved distribution of ions around a biomolecule becomes tractable. We choose carbon atoms as representative sites to characterize ion–residue interactions two bonds away from donor/acceptor atoms for each amino acid (Appendix A). The Bjerrum length [46]— 0.70 nm at 300 K in a physiological solution—represents a meaningful distance at which to compare local ion distributions. Combining this approach with SPEADI’s TRRDF enables the study of ion distributions associated with particular conformational clusters or ion-stabilized states of IDPs. SPEADI’s reliability in characterizing ion distributions increases with the number of simulation frames used for each time window. The interval between frames should be short enough to capture the movement of ions into and out of the hydration spheres surrounding atoms. For a typical simulation of roughly 2400 atoms (excluding the solvent), as in the case of Alpha-Synuclein, this means an interval between 0.1 and 1 ps between frames, depending both on the force field parameters and the simulated ion concentration.

### 2.3. Implementation

Data representing the trajectory of an MD simulation are stored as a sequence of multidimensional arrays. Each array stores the values of a given physical property, e.g., position or velocity, for all particles at a single point in time (the frame). SPEADI’s TRRDF partitions a trajectory in user-specified windows. Additionally, SPEADI is able to account for any simulation cell with periodic boundary conditions, including triclinic and dodecahedral geometries.

SPEADI is released as a Python package that provides functionality to calculate both RDF (Equation (Equation 2)), TRRDF (Equation (Equation 5)) as well as the integral thereof (Equation (Equation 7)) from simulation data in an efficient manner. Additionally, SPEADI provides functionality to calculate the VHF (see Appendix A and references [43,47,48,49,50,51,52]).

All of the implemented functions may be calculated between any multiple groups of reference particles and any multiple groups of target particles. These groups may be of arbitrary size and are defined using the MDTraj [34] selection language. Computation of the functions runs in parallel between all reference and target groups. SPEADI relies on MDTraj as a back end for reading trajectory data and as such, it extends and fits into the MDTraj ecosystem. SPEADI can run interactively in Jupyter Notebooks or Google Colaboratories. It outputs data in Numpy arrays [53,54] that are inter-operable and easy to transform and annotate with, e.g., Xarray [55].

### 2.4. Molecular Dynamics Simulations

Molecular Dynamics (MD) simulations were conducted using the GROMACS software package [35,56]. The simulated protein structures were initialized as water-filled dodecahedral simulation boxes with periodic boundary conditions and minimum distances of 3.5 nm between the boundary and the protein. Sodium chloride salt was used to neutralize the protein and added in excess to obtain a physiological 150 mmol/L^−1^ concentration.

The evaluation of long range electrostatics beyond 1.2 nm was conducted using the Particle-Mesh Ewald (PME) method [57]. The van der Waals interactions were evaluated using a 1.2 nm cutoff. Temperature and pressure conditions were maintained using a Nosé–Hoover thermostat [58,59] together with a Parrinello–Rahman barostat [60]. MD time steps of 2 fs were integrated using the leapfrog algorithm.

The protein systems were first energy-minimized at 0 K, then underwent simulated annealing up to 300 K prior to 1 ns equilibration simulations in the NVT ensemble. The production simulation was then conducted using an NPT ensemble at 300 K.

#### 2.4.1. Alpha-Synuclein

Alpha-Synuclein (AS) in its wild-type and E46K mutant forms was simulated using the Replica Exchange with Solute Tempering (REST2) [61] algorithm implemented in the PLUMED plugins [62,63] for GROMACS, with 32 exchanging replicas at the following temperatures: 300.00, 305.10, 310.28, 315.55, 320.91, 326.38, 331.90, 337.52, 343.22, 349.03, 354.91, 360.90, 366.98, 373.16, 379.44, 385.81, 392.30, 398.88, 405.57, 412.37, 419.26, 426.29, 433.41, 440.66, 448.01, 455.48, 463.09, 470.81, 478.65, 486.61, 494.70 and 500.00 K. These temperatures gave the individual replicas an exchange probability between 30% and 50%, as calculated by the “Temperature Generator for REMD Simulations” by Patriksson and van der Spoel [64].

The most frequent conformation found during previous simulations by Rossetti et al. was used as the starting structure [65]. The structures were N-terminally acetylated to match the physiological form of AS [66,67]. The E46 site was mutated in PyMOL [68]. Simulations were parameterized using the a99SB-*disp* force field [69] and its accompanying modified TIP4P-D water model. A second wild-type simulation was parameterized using the recent DES-Amber force field [70] with the standard TIP4P-D water model.

#### 2.4.2. Humanin

Humanin (HN) in its S14G, d-S7, d-S14, and d-S7,14 mutant forms was simulated using unbiased MD with the help of GROMACS. The starting structures were taken from the first model in the NMR ensemble [71] deposited by Benaki et al. in the Protein Data Bank (PDB, PDB-ID 1Y32) [72,73,74]. Mutations were conducted in PyMOL [68]. The HN was capped by acetyl and N-Methylamine groups at the N- and C-terminus, respectively, to limit its interaction with the ions. HN simulations were all parameterized using the a99SB-*disp* force field and its accompanying water model.

One REST2 MD simulation for wild-type HN was conducted, with 14 exchanging replicas at the following temperatures: 300.00, 312.24, 324.98, 338.19, 351.91, 366.29, 381.10, 396.54, 412.52, 429.10, 446.34, 464.21, 482.81 and 500.00 K. These temperatures gave the individual replicas an exchange probability between 30% and 50%, as calculated by the “Temperature Generator for REMD Simulations” by Patriksson and van der Spoel [64].

## 3. Results

### 3.1. Detection of Mutant Effects in Alpha-Synuclein

We applied TRRDFs using SPEADI to simulations of AS to illustrate the difference in ion interactions both between conformational clusters in a single trajectory and across mutations.

AS is an intrinsically disordered protein with 140 residues known to aggregate into fibrils in so-called *Lewy* bodies found in the tissues of those afflicted by Parkinson’s Disease (PD) [75]. AS may be considered a biomarker for PD [76]. AS oligomers with a β-sheet structure are known to be toxic to cells in vitro [77,78].

AS contains three separate functional domains: the N-terminal domain (residues 1 to 60) that tends to form an α-helix in association with phospholipids [79], the central domain (residues 61 to 95) involved in amyloid binding (Non-Amyloid Component (NAC)) that is one of the non-Amyloid components of Alzheimers’ Disease (AD) plaques [80,81], and the acidic C-terminal domain (residues 96 to 140). The C-terminal domain is highly charged, containing 10 glutamic acid and 5 aspartic acid residues and is highly disordered. The C-terminal domain is thought to interact with and shield the NAC region, inhibiting aggregation unless bound to metal ions or being protonated at a lower pH [82,83,84].

Point mutations identified as being involved in autosomal dominant or familial PD are restricted to the N-terminal domain, in particular involving residues between 46 and 53.

The NAC region contains two of the same conserved KTKGEV sequence motifs that characterize the N-terminal region, yet is thought to be shielded by the negatively charged C-terminal region. The distributions of negatively charged ions around the positively charged nitrogen atoms should differ between the N-terminal and NAC domains [81]. Two pre-NAC regions, P1 (residues 36–42) and P2 (residues 45–57), have recently been identified as critical to the function and the aggregation of AS [85].

A 25 ns MD simulation of wild-type AS in its acetylated wild-type form (AcAS) and a 100 ns MD simulation of E46K-mutated AS (also acetylated) were conducted using REST2 sampling and the a99SB-*disp* force field (see the Appendix A for further details) (we assigned an index of 0 to the acetyl residue to preserve comparison with unacetylated structures). The simulations were conducted in a NaCl solution at physiological concentration (150 mmol/L^−1^). Each simulation was stored twice, using two different modes. One mode where a “wet” trajectory for solvent, ion and protein atoms was stored at 10 ps intervals for the shorter simulation of the wild-type and at 100 ps intervals for the longer simulation of the mutant. The second trajectory was a “dry” trajectory for protein and ion atoms, but stored using an extremely high temporal resolution at 10 fs intervals for both simulations. Both simulations were converged after 12 ns based on the cumulative average of the secondary structure content using the DSSP algorithm [86] (Appendix A).

The converged part of the “wet” trajectories was clustered into 20 K-means clusters using a t-distributed Stochastic Neighbor Embedding (t–SNE) projection [87] of pairwise Root-Mean-Square Displacements (RMSDs), following a procedure recently published by Appadurai et al. [88] (Figure 1D,E; further information in the Appendix A).

The wild-type trajectory shows a bimodal distribution for the protein radius of gyration ($R_{g}$), with peaks centered at 2.5 and 3.4 nm (Figure 1A). The wild-type clusters fall into two broad categories: compact conformations with $R_{g}$ distributions between 2.1 and 2.9 nm, as well as extended conformations with $R_{g}$ distributions between 2.9 and 4.2 nm (Figure 1B). The $R_{g}$ distribution for the E46K mutant trajectory shows only a single broad peak at 3.2 nm (Figure 1A). The separation of the mutant clusters into compact and extended categories is therefore not as clear as for the wild-type. The $R_{g}$ distributions of the mutant clusters are concentrated around 2.9 and 3.3 nm. Overall, the average $R_{g}$ of the wild-type ((3.08 ± 0.49) nm) and mutant ((3.33 ± 0.45) nm) falls within a single standard deviation).

The largest clusters were chosen as representatives for compact ($R_{g}<$
3.1 nm) and extended ($R_{g}\geq$
3.1 nm) categories (Table 1). These clusters also correspond most closely to the peaks and troughs observed in the distributions of $R_{g}$.

Chloride and sodium ion TRRDFs were calculated for each residue’s ion interaction site for both “dry” trajectories using SPEADI. Each 10 ps TRRDF time window was then assigned to the respective conformational cluster (Appendix A).

To compare the ion distributions associated with different conformational states, we use the representative compact and extended conformational structures. The TRRDFs reveal that the chloride and sodium ions’ distribution are complementary: while chloride distributes mostly around the N-terminal region (Figure 2 and Appendix A), sodium prefers the C-terminus (Figure 3 and Appendix A), independently of the system and compactness of the conformations. Despite this general trend, each distribution then features distinct differences when comparing between wild-type and mutant or between compact and extended conformations of each system.

For the overall chloride distribution, peaks are found both at K6 and K10 in all the representative conformations of the wild-type and mutant AS; however, the extended representatives display a higher propensity for chloride ions around the N-terminal region compared to the compact ones. The extended wild-type has higher peaks at D2 and G14 with respect to the extended conformation of the mutant, while the total number of chloride ions for both extended systems is comparable.

Over the NAC region, comparable numbers of chloride ions can be observed for all the analyzed systems, with only the compact wild-type conformation featuring a peak around A76 (Appendix A). Only a few chloride ions can be found at the beginning of the C-terminal region for any of the analyzed conformations in both mutant and wild-type AS.

The overall sodium concentrations around the N-terminus are low for all analyzed systems, with exception of the wild-type compact conformation which features some small peaks in the middle of the N-terminal domain (Figure 3). In the NAC region, the sodium ion concentration is overall higher than in the N-terminal region for all the analyzed systems. The wild-type shows more sodium ions around the NAC, particularly in the second half. The high negative charge of the C-terminal region of course attracts a large number of sodium ions, in all conformations from both wild-type and mutant AS. Compared to the mutant, the total number of sodium ions for the wild-type is significantly higher for both conformations, with high peaks around D121, N122 and E123, as well as E130 and D135.

The summarized results of the ion distributions help to provide insights on residues that play a key role for the function of AS. The N-terminal lysines K6, K10 and K23 displaying the highest chloride concentrations are also the most important ones for AS’s interaction and binding to lipids [90,91]. G47 is the center of a transient β-hairpin thought to be the start the wild-type aggregation pathway [92,93]. A76 plays a complex yet important role in the formation of dimers, oligomers and fibrils [94,95]. D121, N122 and E123 are known binding sites for iron ions that can induce aggregation [96,97]. E130 is the preferred binding site for the aggregation inhibitor spermine, which decreases the negative charge in the C-terminal region [98,99]. All of these interactions correlate with peaks in the ion distributions of either the wild-type compact or extended conformations. Notably, the prominent sodium peaks at D121, E130 and D135 are missing after mutation to lysine at position 46. This should impact the function of the mutant at these residues in any interaction where electrostatics plays a role.

To investigate the correlation between local ion–residue interactions and the overall structure and conformation of AS, we constructed distance matrices for each of the clusters representing the analyzed systems, and overlaid them with the ion distributions (Figure 4). The effects associated with the ion distributions fall into two categories: One category for inter- and a second for intradomain effects. In the following, we define contacts as the mean distance between residues below a 0.5 nm cutoff.

In the first category, chloride ions around the N-terminal domain correlate positively with the interdomain distance between the N- and C-term, i.e., the higher the ion concentration, the higher the interdomain distance. Indeed, the compact conformations of both systems have fewer chloride ions than the extended conformations (Figure 2). Sodium ions show the opposite (negative) correlation, i.e., the higher the ion concentration, the lower the interdomain distance. Interestingly, both ion concentrations around the NAC domain correlate negatively with the NAC-C distance, i.e., both ion distributions favor the formation of interdomain contacts; peaks for both sodium and chloride coincide with contacts between the domains. Finally, the C-terminal region shows no clear correlation between the ion distributions and interdomain distances involving it.

In the second category, the chloride concentration around the whole N-terminal domain correlates positively with the intradomain distances (Appendix A). Accordingly, the extended conformations of both systems have a higher total concentration of chloride ions at the N-terminus than the extended conformations. Perhaps owing to the low amount of sodium ions around the N-terminus, no correlation could be observed between its concentration and intradomain distances. Interestingly, the chloride distribution in the NAC correlates negatively with the intradomain distances, i.e., positively with the number of intradomain contacts (Appendix A). Indeed, the compact conformations of both systems show higher concentrations of chloride compared to the extended conformations. There is no large difference in the overall number of sodium ions at the C-terminus between compact and extended conformations in both systems (Appendix A). Therefore, no clear correlation between the ion concentrations and the intradomain distances could be found.

The results we obtained for the correlation between the ion distributions and the conformation and structure of AS can be summarized as follows: Extended conformations are associated with more chloride ions around the N-terminal region compared to the compact conformations, thus shielding the N-terminus from interaction with the C-terminus. Complementary to chloride, the sodium concentration around the N-terminus correlates negatively with the distance between it and the C-terminal domain. This is in line with the general distributions of chloride and sodium ions observed above—the C-terminal region preferentially attracts sodium, and the sodium ions at the N-terminus may contribute to attracting the C-terminus. This would suggest an electrostatic bridging effect, where sodium ions around negatively charged residues in the N-terminal region are mediating attractive connections to the C-terminus (at a distances greater than those observed for hydrogen bonds or salt bridges). As this was the only effect we observed for the wild-type and not for the mutant, it could suggest a relation to their different aggregation propensities. Accordingly, our radius of gyration analysis shows a higher propensity for the mutation to assume extended conformations with respect to the wild-type (Figure 1). Moreover it could suggest an explanation for an interesting effect in aggregation propensities. The results from experiments and Coarse Grain Molecular Dynamics (CG-MD) show that sodium chloride concentrations of roughly 100 to 150 mmol/L^−1^ stabilize the extended conformations that are thought to lead to the aggregation of AS [100,101,102,103].

### 3.2. Ion Equilibration Depends Strongly on Force Field Parameters

Owing to their large structural heterogeneity and rough but shallow free energy surface, the choice of force field simulation parameters plays an import role in IDP simulations. SPEADI can clearly show the difference in ion distributions due to the force field parameters (Appendix A). Using the DES-Amber force field [70], the distribution of chloride ions around the N-terminal region is significantly increased compared to a99SB-*disp*. The location of the peaks are broadly the same, with a notable exception at K97 in the transition between NAC and C-terminal regions. The mean number of chloride ions around residues with peaks increased by a factor between 2 and 3. In general, lysine residues show a much higher propensity for chloride ions when using the DES-Amber force field. The difference in sodium ion distribution is more nuanced, with a99SB-*disp* showing a slightly higher propensity at D121, around E131, and in the NAC region, whereas DES-Amber shows higher propensities around G101 and N103. It thus becomes clear that the parameters included in DES-Amber do not impact the distribution of sodium around the highly charged C-terminal region, but have a much larger effect on the distribution of chloride ions in the N-terminal region and around the mutation site E46.

SPEADI can also be used to analyze the speed at which the distributions of ions equilibrate during an IDP simulation. This may be done by observing the time it takes for the cumulative average of ion density at a residue to reach the mean density (Appendix A). Overall, the sites with the highest number of ions are also the slowest to equilibrate. DES-Amber shows a much slower equilibration of chloride ions than those parts of the N-terminal region that correspond to positively charged residues. This is due in part to the comparatively high number of chloride ions present on average. a99SB-*disp* shows a much faster equilibration of chloride ions.

These results suggest that the presence of multiple same-charge ions in the hydration shell of an atom represent a rate-limit on the equilibration of ions. Sites with a higher equilibrium number of ions are thus likely to have charges not fully screened during most of the simulation. This may in part explain their propensities for inter- and intramolecular contacts.

Overall, the time-resolved analysis of ion distributions reinforces previous observations that the slow equilibrations of ions are a rate-limiting step during classical IDP simulations [28]. This rate-limit should inform the choice of simulation techniques and parameters. Enhanced conformational sampling of IDPs beyond the sampling of REST2 simulations are being successfully developed in the form of variational autoencoders [104]. We are therefore pursuing the development of methods to speed up the equilibration of ions using machine learning techniques.

### 3.3. Detection of Mutant Effects in Humanin

Next, we applied SPEADI to look for differences in ion distributions caused by mutations in Humanin (HN).

HN is the smallest intrinsically disordered protein found in humans [71]. It has been shown to provide protection against cytotoxic Amyloid-β(Aβ) aggregates as well as to protect against cell damage occurring in the course of hyperglycemia and ischemia [105,106,107,108,109,110]. We use HN as an ideal example to demonstrate the different analysis modes SPEADI provides for IDPs.

Several mutations and d-isomers of HN have been described. Compared to wild-type HN, the S14G mutant (HNG) displays an increased potency against Aβ-induced neuronal cell death in vitro, while the C8A mutant (HNA) shows a complete lack of neuroprotective activity [105,111,112]. d-isomers of both S7 and S14 residues, as well as a combination of both, have been found in samples from patients [113]. Isomerization of the S7 residue leads to decreased neuroprotective activity, while isomerization of the S14 residue leads to an increase in neuroprotective activity similar to that displayed by HNG [113].

The efficacy of HN and its variants correlates with their propensity to form β-sheets [113]. HNG, as well as the d-S14 and d-S7,14 isomers, shows an increased preference for β-sheet conformations when investigated using Circular Dichroism [113].

To investigate the effect of mutation on the ion distribution in HN, we performed 1 μs unbiased MD simulations of wild-type HN, HNG, d-S7, d-S14 and d-S7,14 mutants using the a99SB-*disp* force field (see the Appendix A for details).

We calculated the TRRDFs of both chloride and sodium ions as well as the RDFs. The latter were obtained by averaging the TRRDFs over time. The RDFs and integrals of the average over the whole protein indicate that the mutations have a small effect on the overall number of ions around HN (Appendix A). However, there are a number of local differences (Figure 5).

#### 3.3.1. Local and Allosteric Effects at Positions S7 and S14

In all the systems analyzed, little or no differences in the ion distributions were observed directly at the S7 site in HN (Figure 5 and Figure 6A). However, the S14 site shows large changes in the sodium concentration upon mutation of either the S7 or S14 site and for the double mutant (Figure 5 and Figure 6B). Mutation to glycine or isomerization of S14 increases the local S14 sodium concentration compared to the other mutants, while isomerization of S7 (d-S7 mutant) has the opposite effect. Interestingly, the effect of isomerization of both serines (d-S7,14 mutant) results in an S14 sodium concentration between that of the single isomerizations (d-S7 and d-S14).

#### 3.3.2. Ion-Stabilized Structures Centered on S14

The integral of the sodium RDF at S14 for the wild-type shows a concentration almost twice as high with respect to the mutant with the lowest concentration (d-S7,14, Figure 6B). Upon investigation of the wild-type TRRDF, a strong and constant sodium signal is observed at S14 from 290 to 310 ns simulation time (Figure 7A).

Further investigation of the part of the wild-type trajectory corresponding to the intense sodium signal revealed an ion-stabilized meta-stable structure. This structure formed around a sodium ion coordinated by backbone oxygen at L11, L12, S14 and D17 (Figure 8A). The loop structure is stable for around 20 ns, i.e., for the duration of the sodium ion presence. P19 closes the loop and stays in close contact with L11 throughout this time.

We investigated the trajectories of mutated HN using the TRRDFs (Figure 7B–E) to detect any ion-stabilized structures. Only the d-S7 mutant showed a similarly strong signal in the TRRDF for sodium ions around S14. Here we again found an ion-stabilized structure centered on S14 (Figure 8B). The latter involved the same backbone oxygen as in the wild-type, and was stable for around 10 ns. These stable loop structures were only found in simulations where S14 was not modified. No loops were found at the S7 site.

The presence of this ion-stabilized motif at S14 correlates negatively with the neuroprotective effect of HN against Aβ. Claiming any causal relationship would require further experimental investigation.

To further understand the behavior of such an ion-stabilized structure centered on S14, we enhanced the sampling of the wild-type HN with REST2 simulations (see the methods for details).

Comparison of the wild-type unbiased and REST2 TRRDFs shows a difference in the local behavior of sodium (Appendix A). Overall, in the REST2 trajectory, sodium ions are more consistently present around S14 than in the unbiased trajectory. However, no ion-stabilized structure at S14 (or in any other position) is observed. This is likely caused by the REST2 algorithm that forces the exploration of the conformational landscape through increased kinetic energy. This, in turn, hampers the formation of stable or meta-stable protein–ion complexes. On the other hand, SPEADI also shows that the REST2 algorithm leads to the faster equilibration of ion distributions compared to unbiased MD (Appendix A). This is particularly evident for the residues E15, D17, and K21 described below, where the concentrations of counter-ions are highest. In summary, REST2 MD enhances ion equilibration, but may obscure ion-stabilized structures.

#### 3.3.3. Allosteric Effects of Mutations

The large differences in the ion distributions around the charged residues E15, D17, and K21 (Figure 5) suggest an allosteric effect of mutations through changes in the ion distributions. To further explore the ion dynamics at these positions, we calculated the counter-ion RDFs (Figure 6). The RDFs show peaks that are generally broader compared to the sodium peak at S14. This indicates that counter-ions are free to move into and out of the protein hydration shell at these points. Large peaks in the sodium concentration are observed at both E15 and D17 across all mutants. E15 has been experimentally linked to binding with Aβ42 residues 17 to 28 [114]. D17 was predicted to be the location of a salt bridge between HN and Aβ by structural modeling [114]. The largest peak in the chloride distribution across the mutants is observed at K21, which is the binding site of HN to Insulin-like growth factor-binding protein-3 (IGFBP-3) [115].

Concerning the amount of ions in the E15, D17, and K21 positions discussed above, the S14G mutant is always observed with the highest peaks in both sodium and chloride distributions, while d-S14 shows the second highest peaks. At E15, d-S7 shows a higher concentration of sodium ions compared to the double mutant d-S7,14. This is the opposite at D17. At K21, as at D17, d-S7,14 again shows a higher chloride concentration than d-S7. Interestingly, the order of counter-ion peaks at D17 and K21 of the mutants corresponds exactly to the experimentally reported neuroprotective activity against Aβ [105,111,113]. This suggests that a change in the effective electrostatics around a binding interface plays an important role in at least one of the biological functions of HN.

For the interaction between HN and Aβ, our results suggest the following roles for ions: Sodium ions around E15 and D17 likely contribute to attracting and orienting the binding site in Aβ. Sodium ions also likely shield the binding site in HN (residues 5–15) from the electrostatic charges on E15 and D17. Chloride most likely contributes by shielding the binding site from K21 (and the other basic C-terminal residues). As the mutations with the highest activities against Aβ(S14G and d-S14) simultaneously show the highest concentrations of counter-ions at E15, D17, and K21, the presence of both ions might be equally important.

## 4. Discussion and Conclusions

Ions play an import role in the structure and function of proteins and IDP in general. This is illustrated by the links between ion distributions, functional and mutation sites, conformational states, and transient ion-stabilized structures. TRRDFs preserve information on the dynamics of chemical environments. This allows investigation of changes in simulated ion distributions over time and provides the ability to correlate these with other calculated properties. Care must be taken, however, as these insights depend on the simulation technique and the accuracy of the parameters, including periodic boundary effects and water layer size [116]. TRRDFs are also able to obtain distributions associated with separate conformational states.

Our tool to calculate TRRDFs, SPEADI, is designed take advantage of High Performance Computing (HPC) resources as it can utilize both multi-core/multi-CPU architectures as well as accelerators such as GPUs and TPUs [117,118]. However, it can also take full advantage of resources available on modern desktop computers and laptops (for benchmark comparisons, see Appendix A). This makes it possible to characterize a large number of sites in a biomolecule in parallel.

Overall, we have demonstrated that SPEADI is useful for the characterization of IDP–ion interactions, and may lead to new insights into the dynamics and behavior of ions around IDP surfaces, and thereby ultimately into the functions of IDPs.

SPEADI is open-source and free to use, with several examples included. All RDF and TRRDF results were obtained using version 1.0.0 of SPEADI [119].

## Figures and Tables

**Figure 1 biology-12-00581-f001:**
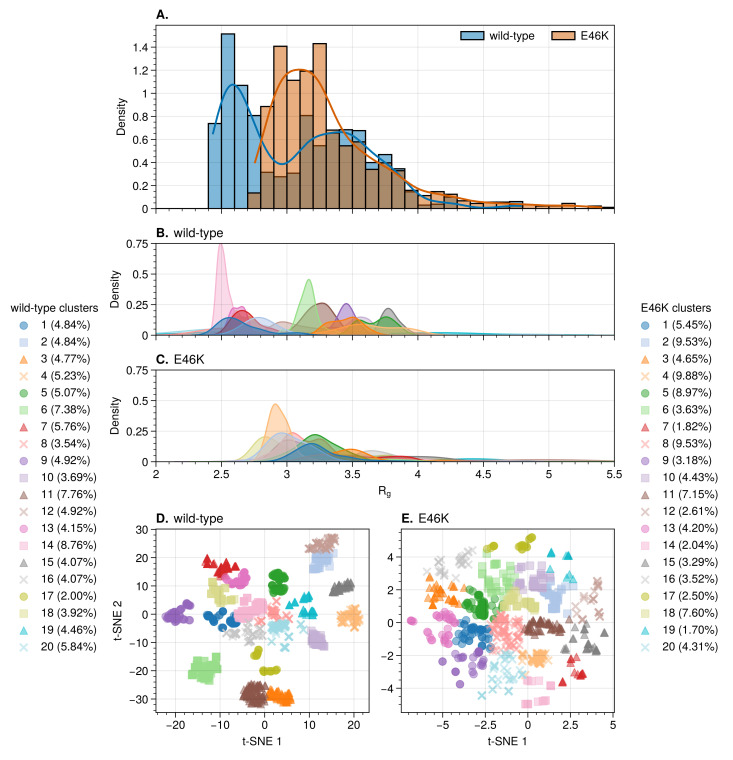
Radius of gyration of wild-type and E46K-mutated AS. Distributions over the converged part of the trajectories (**A**), and over the 20 K-means wild-type (**B**) and E46K mutant (**C**) clusters. The converged t–SNE projections of RMSD used to obtain the K-means clusters of the wild-type (Perplexity = 100) (**D**) and E46K mutant (Perplexity = 350) (**E**).

**Figure 2 biology-12-00581-f002:**
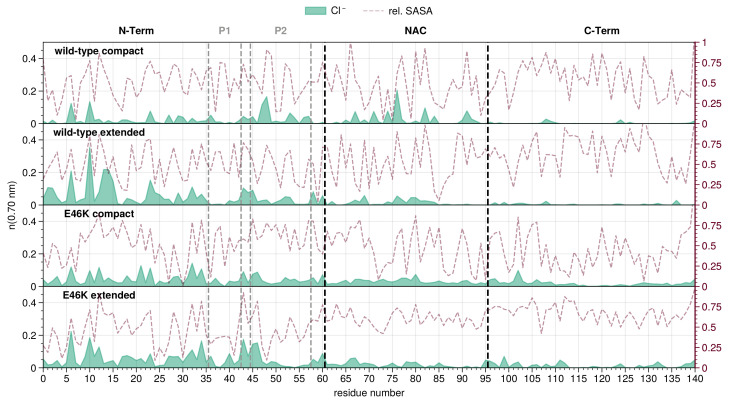
Chloride ion distribution along the residue index for compact and extended conformations of wild-type and E46K-AS. Residue index 0 is the acetylated N-terminus. Mean number of ions (running coordination number) is given as the integral of g(r) for the ion up to the second hydration shell (r≤0.70 nm). Maroon lines indicate the average Relative Solvent Accessible Surface Area (RSA) for each residue (right y-axis) [89].

**Figure 3 biology-12-00581-f003:**
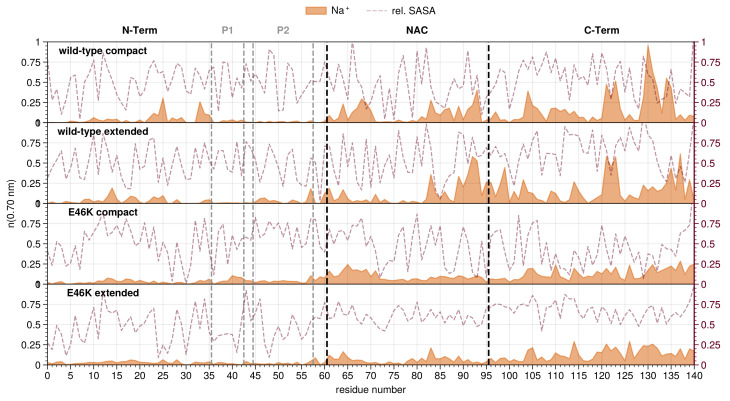
Sodium ion distribution along the residue index for compact and extended conformations of wild-type and E46K-AS. Residue index 0 is the acetylated N-terminus. Mean number of ions (running coordination number) is given as the integral of g(r) for the ion up to the second hydration shell (r≤0.70 nm). Maroon lines indicate the average RSA for each residue (right y-axis) [89].

**Figure 4 biology-12-00581-f004:**
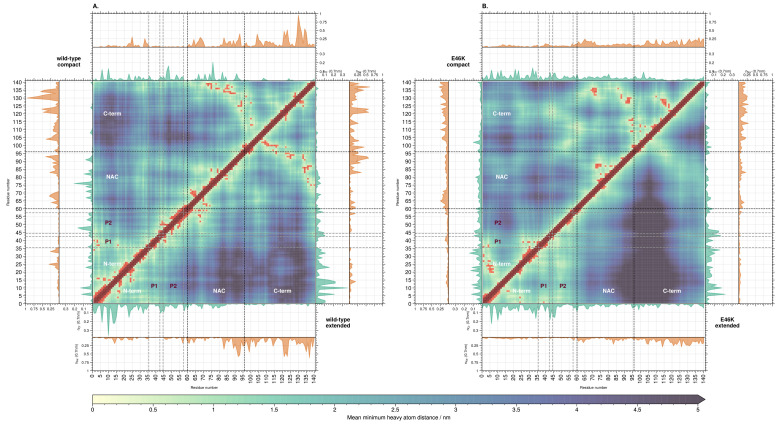
Residue distance map of AS for wild-type compact (**A**, upper triangle) and extended (**A**, lower triangle), as well as E46K-AS compact (**B**, upper triangle) and extended (**B**, lower triangle) conformations. Red points mark contacts (mean distance ≤0.5 nm). Insets show the mean number of chloride (green) and sodium (orange) ions within a distance of 0.70
nm.

**Figure 5 biology-12-00581-f005:**
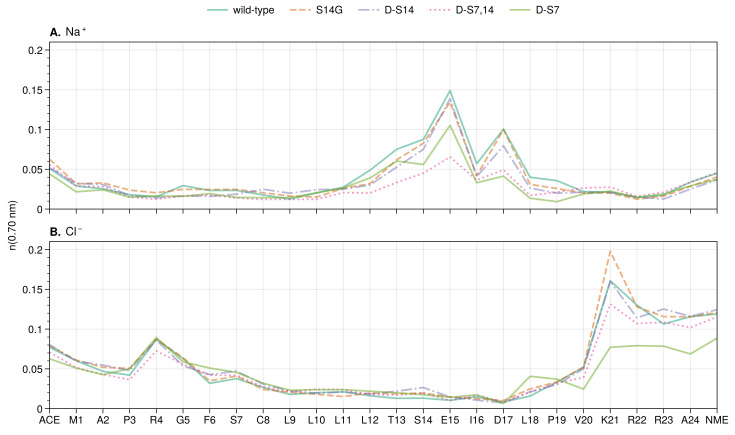
Sodium (**A**) and chloride (**B**) ion distribution along the residue index for wild-type and mutants of HN. Mean number of ions (running coordination number) is given as the integral of g(r) for the ion up to the second hydration shell (r≤0.70 nm).

**Figure 6 biology-12-00581-f006:**
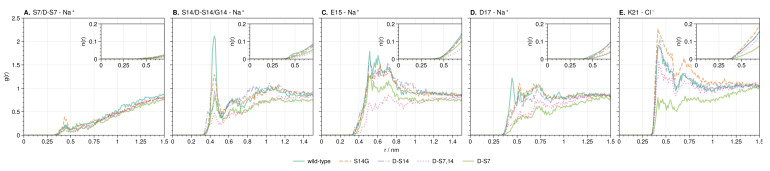
RDF of counter-ions around S7 (**A**), S14 (**B**), E15 (**C**), D17 (**D**) and K21 (**E**) from simulations in wild-type and mutations of HN. Insets show the average number of sodium or chloride ions around each site at 150 mmol/L^−1^ salt concentration.

**Figure 7 biology-12-00581-f007:**
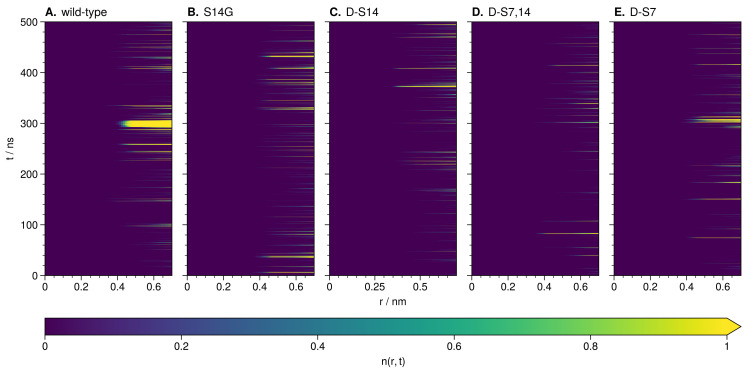
TRRDF for residue 14 during unbiased MD simulations of wild-type (**A**) and mutations (**B**–**E**) of HN. Increasing brightness indicates a higher n(r,t) value (the mean number of sodium ions up to point *r*).

**Figure 8 biology-12-00581-f008:**
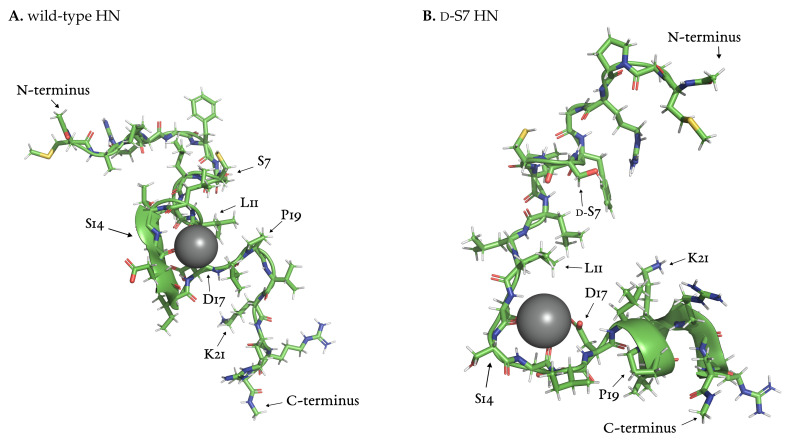
Sodium ion-stabilized structures found in wild-type (**A**) and d-dS7 (**B**) HN, both centered on S14. Snapshots taken at 297.14
ns (**A**) and 306.08
ns (**B**) of simulation time.

**Table 1 biology-12-00581-t001:** Clusters chosen to represent compact and extended conformational structures in trajectories of wild-type AS and E46K mutant.

Trajectory	Conformation	Cluster	% of Converged Trajectory	$R_{g}$/nm
wild-type	compact	14	8.76	2.52 (0.05)
	extended	11	7.76	3.25 (0.11)
E46K	compact	4	9.88	2.95 (0.09)
	extended	16	3.52	3.71 (0.17)

## Data Availability

SPEADI is published under an LGPL open-source license and is publicly available at https://github.com/FZJ-JSC/speadi, accessed on 17 March 2023. The documentation may be found at https://fzj-jsc.github.io/speadi/, accessed on 17 March 2023. Two Google Colaboratory examples are included that show both the basic usage of obtaining time-averaged RDFs, as well as an example showing advanced usage. The trajectory data that support the findings of this study are available from the corresponding author, G.R., upon request.

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
