# Peer review of "SPEADI: Accelerated Analysis of IDP-Ion Interactions from MD-Trajectories"

_biology, 2023, doi:10.3390/biology12040581_

Round 1
Reviewer 1 Report
The authors in the present manuscript report the development of a tool to calculate the time-resolved radial distribution from AA molecular dynamics simulations. The authors calculate the RDF, TRRDF, and VHF for two IDPs and sodium and chloride ions.
The developed tool is useful and interesting for the study of IDPs.
Some comments on points that are not clear to me.
(a) pages 1-2, lines 33-35. I remind the authors that IDPs include IAPP (or amylin) responsible for type 2 diabetes (10.1021/acs.chemrev.0c00981).
Page 2, lines 45-46. RDF in the study of biomolecules has also been used to study the structure of biological membranes (10.1016/j.bpj.2016.05.050) to understand the degree of order of cholesterol-interacting phospholipids.
Last, the manuscript needs to clarify whether SPEADI is specific for calculating RDFs of proteins with ions or can be used considering any atom and its neighbours. If the tool is general, it will help characterize the oligomers of IDPs and cell membranes.
Reviewer 2 Report
Generally, this paper is very well written and prepared with considerable care. It addresses an important topic (influence of ions on the conformational equilibria of intrinsically disordered proteins). The software could be of interest to a number of researchers working on IDP structure.
Specific comments:
Line 128, 129: I do not understand the sentence “SPEADI’s reliability in characterizing ion distributions increases with the number of simulation frames used for each time window”. In molecular dynamics simulations, there is just one frame for each time window! The authors should clarify what they mean.
Line 129, 130: “The interval between frames should be short enough ...”. it would be helpful if the authors could provide an indication what “short enough” means for a typical simulation, or what the likely minimum would be
Line 164: spaces between “300and500 K”. Are the temperatures of the 32 replicas equally spaced in this range?
Line 169: why do the authors use a forcefield that has been optimized for DNA for this study? A brief explanation of why this forcefield was chosen would be interesting. In line 332, it is mentioned that lysines attract a higher number of chloride ions in this forcefield - is this due to this specific forcefield version? Could this be because it has been also optimized for DNA-protein interactions that depend on basic amino acids?
Line 171-180: which mutants were chosen in Humanin? List residue positions and substitutions here
Line 180: again, spaces in “300and500 K”
Line 206, 207: why the differences (25 and 100 ns) in simulation time? Any particular reason for that? These simulation times are rather short (especially in comparison to the humanin simulations presented in the second half of the manuscript). Can the authors justify this?
Line 211, 212: why the different sampling times? Any particular reason for that?
The authors display wildtype and mutant data as two separate figures (Fig. 1 and Fig.2). I think that it would be more helpful and effective to combine the data shown into a larger, single figure that allows a more direct comparison of the two data sets by the reader. If that is done, care should be taken to use the same y-range on the plots for mutant and wildtype (currently, the are slightly different in the tow figures).
Also, the clustering is quite essential here. The cluster assignment for the wildtype structure looks visually convincing (Fig. 1), but the cluster assignments for Fig. 2 look much messier. This obviously reflects some differences in the conformational states, but have the authors considered using other clustering methods (other than K-means) that may give a better result? (I agree that this may not make much difference)
Line 233: Can the authors give an indication how long such a SPEADI analysis will typically take on CPUs and/or GPUs? Hours? Days?
Again, Figure 3 and 4 contain data sets that (in my view) would benefit from being presented together as a single large figure to allow a more direct cross-comparison
Line 285, 286: does the lower ion concentration mean that there is more electrostatic interaction of residues with each other, thus displacing the ions?
Line 376-378: This looks better - long simulations and equal treatment of wildtype and mutant
Line 489: concerning data availability, are the authors also prepared to make their trajectory data available. if yes, that should be mentioned here.
Reviewer 3 Report
The Intrinsically Disordered Proteins (IDPs) have been a challenging target for structural study due to its highly disordered nature resulting in many significantly different unstable conformations. The traditional tools for structure determination hardly work on these IDPs. This manuscript “SPEADI: Accelerated Analysis of IDP-Ion Interactions From MD-Trajectories” developed a tool called SPEADI to calculate Time-Resolved Radial Distribution Function (TRRDFs), characterizing the site-specific ion-residue interactions correlated with conformational and transient ion-stabilized states, illustrated by alpha-Synuclein (AS) and Humanin (HN) cases. The manuscript is clearly written, and the results are solid.
Comments/suggestions:
1. The analysis is based on MD trajectories. For large proteins with significantly different conformations, all atom MD simulation will take very long or infinite time to get these conformations if the starting structure is significant different from the final one. Then the question is: how do the authors decide the starting structures? Are they all biological relevant?
2. Any limitations for this method, such as protein size, solution conditions and so on?
3. Under near-physiological conditions, except for sodium and chloride ions, potassium is also a dominant ion. How about the potassium influence on the IDP structures and functions? Why did the authors not include this part?
4. Line 228, “Overall, the wild-type exhibited a lower average Rg ((3.08 ±0.49) nm) compared to the mutant ((3.33±0.45)nm)”. This conclusion needs to be modified. The wild-type Rg is comparable to mutant considering the errors.
Reviewer 4 Report
The authors implemented the time resolved radial distribution function for studying model IDP and ion interactions in water. This is an interesting study. However, it is very well known that the chosen FF parameters and periodic boundary conditions impact IDP-related simulation outcomes more than ordered proteins. Even though the authors used two different FF parameters, none of them embraced idp-specific force field such as ff14IDPSFF developed by Song et al. Machine learning could have been used for conformational mining. Generative autoencoders provide nice solutions to gain insights that are missed by classical studies.
Please, see Gupta et al. Communications Biology, 2022, 5, 610.
Furthermore, and more important... the authors did not mention the effects of periodic boundary conditions and water layer. It is by now very well known that the chosen water layer size impacts all outcomes (structural and thermodynamic) of classical IDP simulation results.
The authors should at least mention these weaknesses in their conclusion part rather than stating that radial distribution functions can provide insights into the function of IDPs by classical simulation studies. In fact, they report that ion distribution is ff dependent.
